# Interactions between *Kazachstania humilis* Yeast Species and Lactic Acid Bacteria in Sourdough

**DOI:** 10.3390/microorganisms8020240

**Published:** 2020-02-11

**Authors:** Belén Carbonetto, Thibault Nidelet, Stéphane Guezenec, Marc Perez, Diego Segond, Delphine Sicard

**Affiliations:** SPO, University Montpellier, INRAE, Montpellier Supagro, 34060 Montpellier, France; mbcarbonetto@gmail.com (B.C.); thibault.nidelet@inrae.fr (T.N.); marc.perez@inrae.fr (M.P.); diego.segond@inrae.fr (D.S.)

**Keywords:** *Kazachstania**humilis*, *Saccharomyces**cerevisiae*, *Lactobacillus*, sourdough, microbial interactions, bread

## Abstract

Sourdoughs harbor simple microbial communities usually composed of a few prevailing lactic acid bacteria species (LAB) and yeast species. However, yeast and LAB found in sourdough have been described as highly diverse. Even if LAB and yeast associations have been widely documented, the nature of the interactions between them has been poorly described. These interactions define the composition and structure of sourdough communities, and therefore, the characteristics of the final bread product. In this study, the nature of the interactions between strains of two commonly found sourdough yeast species, *Kazachstania humilis* and *Saccharomyces cerevisiae*, and lactic acid bacteria isolated from sourdoughs has been analyzed. Population density analysis showed no evidence of positive interactions, but instead revealed neutral or negative asymmetric interaction outcomes. When in coculture, the yeasts´ population size decreased in the presence of LAB regardless of the strain, while the LAB´s population size was rarely influenced by the presence of yeasts. However, a higher maltose depletion was shown in maltose-negative *K. humilis* and maltose-positive obligately heterofermentative LAB cocultures compared to monocultures. In addition, tested pairs of obligately heterofermentative LAB and *K. humilis* strains leavened dough as much as couples of LAB and *S. cerevisiae* strains, while *K. humilis* strains never leavened dough as much as *S. cerevisiae* when in monoculture. Taken together, our results demonstrate that even if higher fermentation levels with increased maltose depletion were detected for *K. humilis* and obligately heterofermentative LAB pairs, these interactions cannot be ecologically classified as positive, leading us to rethink the established hypothesis of coexistence by facilitation in sourdoughs.

## 1. Introduction

Yeast and bacteria coexist in diverse habitats [1,2,3]. Their coexistence and interactions influence ecosystem function and persistence. In food ecosystems, they are responsible for many different processes including acidification, flavor generation, and ethanol and CO_2_ production, as well as biopreservation [2]. Understanding how bacteria and yeast interact, therefore, not only sheds light on the ecology and mechanisms of their interaction, but also allows us to develop new microbial starters for the food industry.

Among fermented food products, bread is of cultural and historical importance in many countries. Bread can be made by adding yeast starters, but also by using sourdough. The benefits of sourdough have been shown; they include the potential to lower the glycemic index, to increase mineral bioavailability, and to increase the diversity of flavors, among others [4].

Sourdough is a mixture of flour and water where fermentation occurs, leading to the leavening of dough. A microbial community composed of lactic acid bacteria (LAB) and yeast is responsible for the fermentation process [5]. These microbial communities are quite simple, usually harboring one effective yeast species and two effective LAB species [6]. However, a rather high microbial diversity has been described when considering different sourdoughs all over the world; more than 60 LAB species and 30 yeast species have been described so far [7,8,9,10,11,12,13,14,15,16]. Most described LAB species belong to the genus *Lactobacillus*, and most yeast species to the genera *Saccharomyces*, *Kazachstania*, *Wickerhamomyces*, *Torulaspora*, and *Pichia*. The most commonly found *Lactobacillus* species is *Lactobacillus sanfranciscensis* (obligately heterofermentative), but many other *Lactobacillus* species were also detected, such as *Lactobacillus brevis*, *Lactobacillus fermentum*, *Lactobacillus hammesii* (obligately heterofermentative), *Lactobacillus paralimentarius*, *Lactobacillus sakei*, *Lactobacillus kimchi*, *Lactobacillus pentosus*, *Lactobacillus curvatus Lactobacillus plantarum* (facultatively heterofermentative), and less frequently *Lactobacillus salivarius*, *Lactobacillus sakei* (homofermentative) [6,7,8,12,13]. Homofermentative lactobacilli ferment hexose almost entirely to lactic acid by the Embden-Meyerhof pathway. They are not able to ferment pentose and gluconate. Facultatively heterofermentative lactobacilli almost exclusively ferment hexoses to lactic acid by the EMP pathway and possess both aldolase and phosphoketolase activities, and therefore, also ferment pentoses and often gluconate. In the presence of glucose, the phosphogluconate pathway enzymes are repressed. Finally, obligately heterofermentative lactobacilli ferment hexoses by the phosphogluconate pathway yielding lactate, ethanol, acetic acid, and CO_2_. Pentose may be fermented by the phosphogluconate pathway [17,18]. The most widespread yeast species in sourdough are *Saccharomyces cerevisiae*, *Kazachstania humilis* (previously named *Candida humilis*), *Kazachstania exigua*, *Pichia kudriavzevii*, and *Torulaspora delbrueckii* [6,14,16]. These yeast species all ferment hexoses in the absence and presence of oxygen, and thus, convert carbohydrates into ethanol, carbon dioxide, and acetic acid. In the presence of oxygen, they are able to ferment hexoses if enough glucose is present, i.e., they are Crabtree-positive [19]. In addition to producing organic acids and ethanol, LAB and yeast are responsible for bread aroma compounds, and are involved in bread texture and mineral bioavailability.

Sourdough microbial communities can be maintained through backslopping (i.e., the repeated inoculation of water and flour with a former chief sourdough), for more than 30 years [16]. The stability of these communities relies on the association and interaction between community members. However, even if the co-occurrence of LAB and yeast has been systematically detected in sourdoughs, the nature of LAB/yeast interactions has not been well explored so far, and the stability of sourdough is debated [15,20,21,22]. There are many types of interactions which can be determined based on how one species affects the other. In ecology, at least five types of interactions can be defined, (i) a negative/negative interaction, described as competition; (ii) a negative/neutral interaction or *amensalism*; (iii) a negative/positive interaction, described as *predation* or *parasitism*; (iv) a positive/neutral interaction, called *commensalism*; and (v) a positive/positive interaction, known as *mutualism.* (iv) and (v) can also be called *facilitation*.

One of the most common LAB/yeast associations in sourdough is the one between *L*. *sanfranciscensis* and *Kazachstania humilis* [20]. This association was first identified in a sourdough from San Francisco, from which the species *L*. *sanfranciscensis* was first isolated [23]. These two species frequently co-occur in firm sourdoughs that are backslopped regularly at room temperature, traditionally defined as Type 1 sourdough. Type 1 sourdoughs, in which *L*. *sanfranciscensis* occurs, usually harbor less LAB species diversity, suggesting competitiveness of *L*. *sanfranciscensis* towards other bacteria species [6]. However, *L*. *sanfranciscensis* is typically found along with the yeast species *K*. *humilis*. The interaction between these species has been hypothesized to be positive/positive (i.e., mutualistic). The lack of maltose consumption ability in strains of *K*. *humilis* has led to the hypothesis that an association with maltose-positive *L*. *sanfranciscensis* is favored by cross-feeding [24,25,26,27]. Maltose from flour is taken up by maltose-positive *L. sanfranciscensis* cells by a maltose/H+ symport and hydrolyzed into glucose and glucose-1-phosphate by a specific maltose phosphorylase. The latter is further metabolized, whereas glucose can be released into the environment to avoid high intracellular concentrations, mainly under stress conditions. As a consequence, glucose becomes available for fermentation by maltose-negative *K*. *humilis* cells [26,27,28]. Maltose metabolism is not the only mechanism described for yeast growth facilitation. *L*. *sanfranciscensis* cells also release amino acids that can be used by yeast cells or other LAB cells [11,21]. In turn, yeast may also facilitate LAB growth through the release of amino acids [24,29,30], peptides, and vitamins [1,21,30]. In addition, the hydrolysis of sucrose into glucose and fructose by yeast cells can also facilitate LAB growth [29,30]. However, the ability to assimilate hexoses as maltose or fructose by yeast and LAB strains is usually undescribed in sourdough studies. There is still no evidence that yeast-LAB co-occurrence in sourdough is actually governed by hexose cross-feeding, even for the *K*. *humilis* and *L*. *sanfranciscensis* pair. Moreover, there is evidence of other mechanisms in yeast-LAB interactions that may be of greater importance. Yeasts and LAB produce and release compounds which modify the physicochemical properties in the environment with either positive or negative effects on the community members. For example, the lactic acid produced by LAB during fermentation can notably exclude or strongly inhibit the growth of other microorganisms by decreasing the pH of the environment. This usually promotes the growth of low-pH adapted yeast and LAB strains compared to non-fermenting species.

This study aimed to analyze the nature of the interactions between *K*. *humilis* and diverse LAB strains found in sourdoughs. Species and strains were chosen in order to present the metabolic diversity found in sourdoughs. Maltose-Negative *Kazachstania humilis* strains, as well as a maltose-positive strain, of *Saccharomyces cerevisiae* were selected. The later possesses a glucose repression system which favors glucose fermentation over maltose fermentation when sufficient glucose is present. This system was hypothesized to limit competition for maltose between maltose-positive LAB and maltose-positive yeast strains. The chosen LAB strains belong to *L. sanfranciscensis*, *L. hammessii* (obligately heterofermentative), *L*. *pentosus*, *L*. *kimchii*, *L*. *sakei*, and *L*. *curvatus* (facultatively heterofermentative), to cover part of the diversity of species present in European sourdoughs [12,13,25]. Obligately heterofermentative LAB have the ability to produce CO_2_, ethanol, lactate, and acetate, and can also use co-substrates such as oxygen or fructose as electron acceptors, leading to the production of mannitol. The reduction of fructose to mannitol regenerates NAD^+^ and pushes fermentation to acetate instead of ethanol or lactate.

The nature of LAB and yeasts interactions was defined using ecological concepts as follows: the effect of strain A on strain B was regarded as positive if the population density of B increased in the presence of A, and as negative if it decreased. It can also be neutral when the population density of A is unaffected by the presence of B. We then compared the growth of the selected yeast and LAB strains in monoculture and in coculture in laboratory wheat sourdoughs. Moreover, we studied fermentation by measuring dough leavening capacity and the concentration of the main carbohydrates (glucose, maltose, fructose), organic acids (lactate, acetate, succinate), and alcohols (ethanol, glycerol, mannitol) in sourdoughs to gain deeper insights into the relationship between carbohydrate metabolism and the nature of the interactions.

## 2. Materials and Methods

### 2.1. Strains and Inocula

Strains used in this study were selected from a large collection of strains previously isolated from French sourdoughs [13]. These strains were taxonomically identified and characterized [9,14]. They were chosen to represent the known metabolic diversity of yeasts and LAB in sourdoughs, and also to have different geographical origins (Table 1, Appendix A). Strain names were encoded as follows: a first character indicating whether if it is bacteria (b) or yeast (y), followed by the sourdough ID (i.e., bakery of origin; Bi), and a final three-letter code to indicate LAB species when suitable. Eight *Lactobacillius* strains were selected, among which three presented obligately heterofermentative metabolism (bB4-sf, bB5-sf, and bB5-ham) and five had facultatively heterofermentative metabolism (bB16-cur, bB5-pen, bB5-kim1, bB5-kim6, and bB4-sak). These strains also harbor different abilities to assimilate substrates (Appendix A). Regarding yeasts, three maltose-negative *K. humilis* strains (yB6-15, yB5-TP1, and yB5-AC1) and one maltose-positive *S*. *cerevisiae* strain (yB10F-9) were selected (Appendix A). Overall, they were isolated from five bakeries (B4, B5, B6, B10, and B16) located in different regions all over France (i.e., Pays de Loire, Ile de France, Champagne Ardennes, Provence-Alpes Côte d’Azur, and Ile de France, respectively). The bread-making practices used in these bakeries were previously described in Urien et al., 2019 [14] and Michel et al., 2019 [16]. Two obligately heterofermentative LAB strains (bB5-sf and bB5-ham), three facultatively heterofermentative LAB strains (bB5-pen, bB5-kim1, and bB5-kim6), and two *K. humilis* strains (yB5-TP1 and yB5-AC1) were isolated from the same sourdough (i.e., B5), and therefore, are likely to have coevolved. LAB strains were kept in our collection at −80 °C in Man-Rogosa-Sharpe-5 (MRS-5) medium with 15% (*v/v*) glycerol, and yeast strains were kept in YD (4 g/L Yeast Extract, 8 g/L glucose) with 20% (*v/v*) glycerol. Before being used as sourdough starter/inoculum, each LAB strain was propagated in MRS-5 medium plates containing 1.5% agar and incubated at 28 °C for 24 h in anaerobiosis. Yeast strains were propagated in YD medium plates containing 1.5% agar and incubated at 28 °C for 24 h.

For inoculation of dough LAB strains were grown for 24 h in liquid MRS-5 medium in anaerobic conditions. Yeast strains were grown for 24 h in liquid YD medium with agitation (200 rpm). Live cells were counted using a flow cytometer (Accuri C6). The starter dough inoculum was defined as 10^7^ cells/g of dough for both LAB and yeast strains. The appropriate volume of cells was pelleted by centrifugation (3500 g for 5 min at 4 °C in Thermo Scientific centrifuge) and resuspended in 2.5 mL or 5 mL of a sterile solution of tryptone and NaCl (1 g of tryptone and 8.5 g of NaCl in 1000 mL of distilled water).

### 2.2. Laboratory Sourdough Preparation and Indirect Fermentation Record

Dough was prepared by mixing 1000 g of wheat flour (sterilized by gamma radiation, Ionisos) and 700 mL of sterilized water using a Kenwood blender/mixer. For each experimental unit, 85 g dough was used. Each experimental unit was then inoculated with 10^7^ cells of LAB and/or 10^7^ cells of yeast per gram of dough in 5 ml final volume of tryptone/NaCl solution. Treatments were defined as monoculture when inoculating with a single strain, or as coculture when inoculating with one LAB strain and one yeast strain. Twenty-Five grams of inoculated dough were then placed in a dough height-measuring device (Chopin Technologies) to measure leavening capacity (i.e., to indirectly measure fermentation). The dough experimental units were incubated at 28 °C, and dough height was recorded every hour for 6 h. These experimental units were kept at 28 °C for 24 h in total. The spare 60 g of dough was also incubated at 28 °C for 6 h and stored at −20 °C until further analysis. Experiments were replicated between three and seven times for each treatment. Samples before inoculation were taken from each dough batch as controls for contamination. Control samples were also stored for metabolite analysis to elucidate dough characteristics before inoculation (more details in the ‘Metabolite Analysis’ section).

### 2.3. Sampling and CFU Count

Two grams of dough per sample were taken at 24 h and resuspended in 40 ml of tryptone/NaCl solution for CFU counts. Samples were vortexed for 1 min for homogenization. Dilutions and plating were done using a spiral plater (Easy spiral, Intersciences, Saint nom, France). For LAB, CFU counts 1 × 10^5^ dilutions were plated on MRS-5 medium containing 1.5% agar and 0.01% (*v/v*) cycloheximide and incubated at 28 °C for 24 h in anaerobiosis. For yeast CFU counts, 1 × 10^4^ dilutions were plated on YPD medium containing 0.01% (*v/v*) chloramphenicol and incubated at 28 °C for 24 h. CFU counts were recorded using an automatic colony counter (Scan 500, Interscience).

### 2.4. Metabolite Analysis

Concentrations of maltose, glucose, fructose, glycerol, ethanol, mannitol, acetate, lactate, and succinate in dough after 6 h incubation were determined by high performance liquid chromatography. The methods used are described in Michel et al. (2016) [12]. In brief, 3 g of dough was diluted in 3 ml of distilled water and vortexed. In order to remove proteins, 250 μl of Carrez A reagent (3.6% (*w/v*) of K_4_(Fe(CN)_6_)3H_2_O) and 250 mL of Carrez B reagent (7.2% (*w/v*) of ZnSO_4_7H_2_O) were added. After filtration (0.22 micron) and centrifugation (16,060× *g* for 15 min at room temperature), the supernatant (water-soluble extract) was then diluted with 10 mM H_2_SO_4_ and analyzed by liquid chromatography using an HPLC system. Compounds were separated on a Phenomenex Rezex™ ROA-Organic Acid H+ column using a simple isocratic HPLC method and identified by retention times with ultraviolet and refractive index detectors. Metabolite concentrations are expressed as g/kg of dough.

The leavening and concentration of metabolites were measured at 6 h of incubation at 28 °C, since these are the time and temperature used in artisanal sourdough baking [14,34]. However, we chose to measure CFUs after 24 h because we were interested in studying stable interactions between isolates that could survive the fermentation process and remain present in dough for the next bread production batch, usually performed one day later. Dough pH was also measured at 24 h.

### 2.5. Acidity Tolerance Analysis

To determine the effect of pH, lactate, and acetate on yeast growth, several media were prepared: (i) YPD with pH ranging from 3.5 to 6 with a step of 0.5, (ii) YPD (pH 4) containing 0/50/100/150/200/250 mM lactate, and (iii) YPD (pH 4) containing 0/35/70/105/140/175 mM acetate. pH was adjusted using 6N HCl or 8N NaOH. Overnight cultures of the four yeast strains were inoculated in triplicate at OD_610_ of ca. 0.005 into 150 μl of these media. Yeast cells were incubated at 28 °C under continuous shaking in 96-wells microtiter plate, and growth was measured for 24 h at OD_610_ using a TECAN Infinite M200 Microplate Reader (TECAN, Männedorf, Switzerland).

### 2.6. Statistical Analysis

The datasets are available under the DOI 10.5281/zenodo.3634462, under a creative commons attribution.

#### 2.6.1. Repeatability

The described experiments were carried out in two batches. The first was carried out in September 2016 and the second in November 2016. Two yeast strains (yB6-15 and yB5-TP1) and two LAB strains (bB5-sf, bB5-ham) were included in both batches and were used to test batch effects using the following model:Y_ijk_ = μ + α_i_ + β_j_ + γ_ij_ + ε_ijk_
where Y_ijk_ is the CFUs/g of sourdough after 24 h of incubation (named as CFUs from now on in the text), α_i_ is the fixed batch effect, β_j_ is the fixed strain effect, and γ_ij_ the interaction effect. No batch effect was found either for yeasts CFUs (F_1,8_ = 1.07, *p* = 0.33) or LAB CFUs (F_1,6_ = 0.09, *p* = 0.77). No interaction effect between batch and strains was found (yeast F_1,8_ = 0.0006, *p* = 0.98: bacteria CFU F_1,6_ = 0.08, *p* = 0.79).

#### 2.6.2. Single Strain Analysis

Since we did not detect batch effects on the CFUs variable, we grouped both batches to test differences in population density in monoculture between strains using a one way ANOVA test, with strains as a fixed effect:Y_ij_ = μ + α_i_ + ε_ij_
where Y_ij_ is the CFUs/g of sourdough after 24 h of fermentation, α_i_ is the fixed strain effect, and ε_ij_ is the residual error. The analysis was done separately on yeasts and bacteria.

For the analysis of dough height, as well as carbohydrates (maltose, glucose, fructose), alcohols (glycerol, ethanol, mannitol), and organic acid (succinate, pyruvate, acetate, lactate) concentration, the following model was used:Y_ijkl_ = μ + α_i_ + β_j_ + γ_jk_ + ε_ijk_
where Y_ijkl_ is either dough height or the metabolite concentration in mg/g of sourdough after 6 h of fermentation, α_i_ is the fixed batch effect, β_j_ is the fixed microbial type effect (bacteria or yeast), γ_jk_ is the fixed strain effect within microbial type (bacteria or yeast), and ε_ijk_ is the residual error.

#### 2.6.3. Interaction Analysis

The effect of each bacterial strain on each yeast strain population density, or inversely, the effect of yeast on bacteria, was tested by comparing yeast-bacteria in cocultures and yeast monoculture, or by inversely by comparing yeast-bacteria in cocultures and LAB monoculture using the following model:Y_ij_ = μ + α_i_ + ε_ij_
where Y_ij_ is the CFUs/g of sourdough after 24 h, α_i_ is the fixed culture effect (mono and cocultures), and ε_ij_ is the residual error. This model was carried out for each yeast strain (yeast model) and for each LAB strain separately (LAB model). For the yeast model, we compared the yeast strain monoculture and all the cocultures with LAB strains (i = 3 to a maximum of 7), while for the LAB model, we compared the LAB strain monoculture and all cocultures with yeast strains (i = 2–5). A complete model including all yeast and LAB strains was not appropriate because the experimental design was not comprehensive enough to test all yeast or LAB strains against each other. In other words, impossible combinations of yeast and LAB strains were grown in coculture in this study.

The effect of yeast-bacteria coculture on metabolite concentration or dough height was tested using the following model:Y_ij_ = μ + α_i_ + ε_ij_
where α_i_ is the fixed culture effect (i = 1–4) and ε_ij_ is the residual error. For each bacteria/yeast strain combination, we compared the metabolite concentration in the noninoculated dough, the LAB monoculture, the yeasts monoculture, and the yeast/LAB strains coculture. This model was carried out for each yeast/LAB combination separately.

FDR was used to account for multiple testing.

#### 2.6.4. Acidity Tolerance Analysis

For the analysis of the pH, acetate, and lactate sensitivity of yeast strains, the following model was used:Y_ijkl_ = μ + α_i_ + β_j_ + γ_jk_ + ε_ijk_
where Y_ijkl_ is OD after 24 h of YPD growth, α_i_ is the fixed effect of pH or Lactate or Acetate concentrations, β_j_ is the fixed strain effect (yeast), γ_jk_ is the interaction effect, and ε_ijk_ is the residual error.

#### 2.6.5. Multivariate Analysis

Principal component analysis was done using R package ‘stats’. Plots were done using R package ‘ggplot2’. Permanova was calculated with the ‘vegan’ package. R version 3.4.2 was used.

## 3. Results

### 3.1. Phenotypic Variation between Strains in Monoculture

#### 3.1.1. Population Density

We first characterized the population size of strains grown in monoculture. The results showed that for every tested strain, either yeast or LAB, population density increased after 24 h incubation. Even though LAB and yeast strains were inoculated at the same initial density (10^7^ cells/g of dough), population density after 24 h was lower on average for yeast strains than LAB strains. It reached 1.6 × 10^8^ CFUs/gr ±0.4 × 10^8^ SE and 1.5 × 10^9^ CFUs/gr ±0.74 × 10^9^ SE for yeast and bacteria respectively on average. Population density also varied among yeast strains (F_3,14_ = 7.38, *p* = 0.003) and among LAB strains (F_7,20_ = 10.19, *p* < 0.001) (Appendix A). The results showed that the *S. cerevisiae* yeast strain yB10F-9 reached a significantly higher population density compared to *K. humilis* strains (Figure 1A). As regards LAB strains, bB5-ham reached a higher population density at 24 h, while bB4-sak showed the lowest population density (Figure 1B).

#### 3.1.2. Dough Height

Changes in dough height in monoculture were analyzed for yeast and only for obligate heterofermentative LAB strains (bB5-ham, b-B4sf and bB5-sf), as facultatively heterofermentative LAB strains are not expected to produce carbon dioxide, and thus, to leaven dough. Measurement of dough height after six hours of incubation showed significant differences between yeast and obligate heterofermentative bacteria (F_1,22_ = 167.3, *p* < 0.01). As expected, obligately heterofermentative LAB strains leavened dough less than yeast strains (0.53 ± 0.2 cm height increased for LAB strains compared to 1.87 ± 0.6 cm for yeast strains on average). This difference was significant after 6 h of fermentation (Appendix A). Among yeast strains, there was no difference in leavening between *S. cerevisiae* yB10F-9 and *K. humilis* strains yB5-AC1 and yB6-15, but they all differed significantly from the *K. humilis* strain yB5-TP1, which showed the lowest dough height at 6 h (see Appendix A, Figure 2, and Appendix A). After 6 h, leavening for yeast yB5-TP1 strain was not significantly different than that for LAB strain bB5-sf, with both being significantly higher than for LAB strains B4-sf and bB5-ham (see Figure 2 and Appendix A).

#### 3.1.3. Concentration of Metabolites in Dough

The levels of the most common carbohydrates, organic acids, glycerol, mannitol, and ethanol in dough were measured at 6 h incubations in monoculture (Figure 3, Appendix A). Again, only yeasts and obligately heterofermentative LAB strains were analyzed.

Results showed that maltose levels significantly decreased when dough was inoculated with the maltose-positive *S. cerevisiae* strain yB10F-9, but not when inoculated with other yeast strains (Figure 3).

Glucose levels decreased significantly for the three maltose-negative yeast strains, but not for the maltose-positive yB10F-9 strain, where glucose levels increased. The glucose level also increased significantly in dough inoculated with obligately heterofermentative LAB strains. Fructose levels decreased for all obligately heterofermentative LAB and yeast strains. As expected, obligately heterofermentative bacteria produced mannitol, while yeast did not. Ethanol levels increased in yeast and obligately heterofermentative LAB monocultures, and showed the highest level in dough inoculated with *S. cerevisiae* yB10F-9, followed by dough inoculated with *K. humilis,* and then by dough inoculated with obligately heterofermentative LAB.

Metabolite balance for alcoholic fermentation was in agreement with the expectations for maltose-positive LAB strains and maltose-negative *Kazachstania* strains. For the obligately heterofermentative LAB strains, the ethanol concentration measured at 6 h of fermentation was significantly correlated to the ethanol concentration predicted from the consumption of maltose and fructose molecules (R^2^ = 0.87, *p* = 0.0007). For the maltose-negative yeast strains (i.e., the *K. humilis* strains), the ethanol concentration was significantly correlated to the predicted concentration of ethanol molecules produced from glucose and fructose (R^2^ = 0.62, *p* = 0.0002).

Glycerol levels significantly increased in dough inoculated with yeast strains, but decreased in dough inoculated with obligately heterofermentative LAB strains. In addition, acetate significantly increased in yeast and LAB monocultures, although obligately heterofermentative LAB produced four times more acetate than yeasts. Lactate levels significantly increased in all obligately heterofermentative LAB monocultures, but did not change in yeast monocultures. Succinate concentration was higher in yeast than in obligately heterofermentative LAB monoculture. Changes in pyruvate levels were not detected for any strain in monoculture.

Among *K. humilis*, different metabolic patterns were also detected (Figure 3). The yeast strain yB5-TP1 produces less glycerol and ethanol than the other *K. humilis* strains, but produces more succinate, suggesting that a higher part of pyruvate goes to the TCA cycle. The obligately heterofermentative bB5-sf strain produces more acetate and lactate and less pyruvate than the other tested obligately heterofermentative LAB strains (Figure 3). In addition, it depletes more fructose and produces more mannitol.

### 3.2. LAB Effect on Yeast

In order to gain a deeper insight into how the interaction with LAB affects yeasts, we compared every variable measured between monoculture and coculture treatments for each yeast strain. Obligately but also facultatively heterofermentative LAB strains were included in this analysis to get a broader picture of the LAB effect on yeasts.

#### 3.2.1. Population Density

Every yeast strain showed significantly lower population density in coculture with LAB strains than in monoculture, regardless of the combination (Figure 4A, Appendix A, *p* < 0.001). In coculture with LAB, yeast population density reached 0.9 × 10^8^ CFUs/g on average, and thus, decreased by a factor of 1.87 on average. For a given yeast strain, the yeast population density in coculture was not significantly different according to the LAB strain inoculated, except for yB6-15/bB5-pen and yB6-15/bB5-kim6, that had a lower yeast population density than yB6-15/bB4-sak. Overall, there were no significant differences of the LAB/yeast competition effect between yeast strains (Appendix A). For both maltose-positive and -negative yeasts, LAB presence resulted in lower yeast population density.

In order to test whether the negative effect of LAB on yeasts could be related to yeast acidity sensitivity, yeasts CFUs were correlated with dough pH after 24 h of fermentation. No significant correlation was detected (R^2^ = 0.04, *p* = 0.48). In addition, yeast sensitivity to a range of pH, lactate, and acetate variation in YPD liquid media was analyzed. pH showed a significant effect on yeast population size (F_1,76_ = 35.23, *p* < 0.0001); the effect depended on the strain (interaction effect, F_3,76_ = 3.73, *p* = 0.015). *S. cerevisiae* yB10F-9 population size after 24 h (as measured by optical density) decreased at pH 3.5, while the population size of *K. humilis* strains decreased at pH 4 and below, indicating that the *S. cerevisiae* yB10F-9 better tolerates a decrease of pH than *K. humilis* strains (Appendix A). However, 28 out of 48 doughs inoculated with cocultures of *K. humilis* strains (58%) showed pH > 4. Therefore, the observed decrease of *K. humilis* population size in the presence of LAB strains could not be explained by sensitivity to pH for every case. At pH 4, the concentration of lactate (tested from 50 to 250 mM) had no significant effect on yeast population size in YPD media (lactate effect, F_1,64_ = 0.22, *p* = 0.6; Appendix A). By contrast, the concentration of acetate had a significant effect (acetate effect, F_1,73_ = 250.9, *p* < 0.001). The population size of *S. cerevisiae* yB10F-9 decreased at acetate 105 mM, while *K. humilis* strain population size decreased at 140 or 175 mM of acetate (Appendix A). These concentrations are far above the concentration of acetate observed in doughs in this study, which were below 20 mM.

#### 3.2.2. Dough Height

We compared fermentation performance between mono and cocultures of the same yeast strain with different LAB strains to get an estimate of CO_2_ production (Appendix A). Both obligately and facultatively heterofermentative strains were included in this analysis. Cocultures of LAB with *S. cerevisiae* yB10F-9 strain showed no difference in dough height when compared with yB10F-9 monoculture. By contrast, *K. humilis* strains showed different dough height after 6 h when incubated in coculture with LAB strains, in some cases. The effect of LAB strains on dough height was either positive or negative, depending on the combination of strains. Obligately heterofermentative strains (bB4-sf, bB5-ham, bB5-sf) showed a positive or no significant effect on dough height. In contrast, facultatively heterofermentative bacteria led to a negative or no significant effect (Appendix A, *p* < 0.001).

### 3.3. Yeast Effect on LAB

#### 3.3.1. Population Density

The yeast effect on LAB was analyzed on obligately and factitively heterofermentative LAB strains. The growth of LAB strains was only rarely affected by the presence of yeasts (Figure 4B). In coculture with yeasts, LAB population density reached 1.31 × 10^9^ CFUs/gr on average, a value equivalent to the monoculture population density. Of the 16 tested pairs, only three showed a decrease of LAB growth in the presence of yeast strains (bB5-ham/yB10F-9, bB5-kim1/yB5-AC1, and bB5-kim6/yB6-15) (Appendix A, *p* < 0.001).

#### 3.3.2. Dough Height

Dough height in LAB monoculture treatments was measured for obligately heterofermentative LAB strains only (bB5-ham, b-B4sf, and bB5-sf). Leavening after 6 h was higher in coculture than monoculture for all tested combinations, reaching the lowest height when incubated with yB5-TP1 strain (Appendix A). The highest dough height was reached in cocultures with the *S. cerevisiae* strain as well as in the pairs of cocultures with *K. humilis*: yB5-AC1/bB5-sf and yB6-15/bB4-sf (Appendix A). In other words, *K. humilis* yB5-AC1 and yB6-15 strains were rising dough as much as *S. cerevisiae* when cocultured with the tested obligately heterofermentative *L. sanfranciscensis* strains.

### 3.4. Metabolite Profiles and pH in Cocultures and Monocultures

The concentration of metabolites in dough fermented with obligately heterofermentative LAB and yeast strains pairs was compared with that of metabolites in monoculture. The levels of acetate were higher in dough containing LAB alone or in coculture with yeast compared to dough inoculated by yeasts alone (Appendix A). No significant changes were observed for acetate concentration between cocultures and obligately heterofermentative LAB monocultures. As expected, lactate levels were higher in cultures inoculated with LAB than in yeast monocultures. Moreover, lactate was lower in cocultures bB5-sf/yB6-15 and bB5-sf/yB10F-9 compared to bB5-sf monoculture, showing a negative effect of yeasts on the lactic acid production by bB5-sf. Pyruvate levels did not show differences between tested LAB monocultures and cocultures. Pyruvate concentration was low but significantly higher in cocultures than in yeast monocultures. The highest pyruvate levels were observed for cocultures with facultatively heterofermentative LAB.

Maltose was not significantly depleted in LAB monocultures. The level of maltose in *K. humilis* monocultures was significantly increased compared to noninoculated dough, suggesting yeasts may release maltose through their amylase activity. Maltose concentration was significantly lower in *K. humilis*/obligately heterofermentative LAB cocultures as compared to yeast monocultures (Appendix A, Figure 5). The level of maltose depleted in these cocultures was higher than that depleted in LAB monocultures. In addition, higher glucose levels were observed in cocultures of *K. humillis* with obligately heterofermentative LAB strains compared to yeast monocultures (Appendix A). This pattern was not observed for the maltose-positive *S. cerevisiae* strain (yB10F-9), where maltose levels decreased similarly in cocultures and yeast monocultures compared to noninoculated dough.

The pattern of carbohydrates depletion and the pH also depends to some extent to the interacting couples. A more efficient fructose consumption and production of mannitol in cocultures pairs of bB5-sf strain with yB5-AC1, yB5-TP1, and yB10F-9 was observed compared to other pairs of obligately heterofermentative LAB and yeasts (Appendix A). Moreover, the bB5-sf/yB5-AC1 pair showed lower fructose levels than bB5-sf monocultures, but this did not translate into higher mannitol levels (Appendix A). As expected, pH in dough mostly depended on LAB metabolism and was lower in experimental units inoculated with obligately heterofermentative LAB than facultatively heterofermentative LAB. The cocultures pairs of bB5-sf with yB6-15, yB5-TP1, and yB10F-9 showed the lowest pH.

### 3.5. Multivariate Analysis

In order to integrate the results obtained for the metabolic profiles and fermentation of dough, a PCA analysis was performed. The first axis, explaining 36.81% of the variation, mostly separated dough according to the metabolism of LAB (i.e., obligately or facultatively heterofermentative LAB), and was related to maltose, pH, succinate, acetate, and mannitol. The second axis, explaining 27.04% of the variation, mostly separated dough according to the presence/absence of yeasts, and was mainly related to ethanol, dough height, and glycerol content (Figure 6). A permanova test revealed that experimental dough units can be separated into six groups, depending on the metabolic characteristics of the inoculated strains (*p* < 0.05, based on Euclidean distance matrix). One group was composed of dough units inoculated with monoculture of the *S. cerevisiae* maltose-positive strain, a second group contained doughs inoculated with *S. cerevisiae* maltose-positive strain and obligately heterofermentative LAB strains, a third group was composed of dough inoculated with monoculture of *K. humilis* maltose-negative strains, a fourth group was composed of dough inoculated with *K. humilis* maltose-negative yeast and facultatively heterofermentative LAB strains, a fifth group was composed of dough inoculated with *K. humilis* maltose-negative yeast strains and obligately heterofermentative LAB, and finally, a sixth group was composed of obligately heterofermentative LAB monocultures.

## 4. Discussion

Sourdough microbial communities are simple and stable, usually harboring one prevailing yeast species and one or two prevailing LAB species [7,15]. While the species diversity of sourdough LAB and yeast have been well documented, there is less information of the intraspecific diversity among strains and on the functional outcomes of different strains pairs. In this study, an analysis of the fermentation patterns in dough revealed metabolic diversity between bacteria strains, between yeast strains, and between yeast/LAB pairs. *K. humilis* strains differ from the *S. cerevisiae* by their higher sensitivity to low pH and their higher tolerance to acetate. Interestingly, *K. humilis* strains yB5-AC1 and yB6-15 were shown to leaven dough as high as the tested *S. cerevisiae* strain. In addition, LAB strain bB5-sf leavened dough as high as yeast yB5-TP1. This LAB strain in coculture with some yeast strains also consumes more fructose and produces more mannitol than other LAB/yeast pairs. These results confirmed the potential of *K. humilis* and obligately heterofermentative LAB strains as good starters for bread production.

Community stability is known to be a consequence of biotic and abiotic interactions. The general cooccurrence of usually one or two prevailing LAB species and one prevailing yeast species led to a common hypothesis of cooperation between LAB and yeasts in sourdough, particularly to the maltose-negative *K. humilis* and maltose-positive *L. sanfranciscensis* facilitation by cross-feeding hypothesis. Here, the interactions between LAB and yeast strains isolated from sourdoughs were analyzed and the aforementioned hypothesis was indirectly tested. A dough media was used to mimic in situ bakery’s conditions. No evidence of positive ecological outcome was found. Instead, it was shown that yeast growth was hampered by LAB presence, while LAB seems unaffected by the presence of yeast. These yeast/LAB interactions can thus be classified as negative or neutral, based on the ecological definition of antagonist/negative interaction. A positive ecological effect of a maltose-negative *K. exigua* on population density of the obligately heterofermentative *L. brevis* species was previously found in synthetic media with wheat-flour hydrolysate [30]. Controversially, this result was not confirmed by this study. Here, three maltose-negative *K. humillis* strains (yB5-TP1, yB5-AC1 and yB6-15) and one *S. cerevisiae* maltose-positive strain (yB10-F9) were included in the study. A negative impact on yeast population density for every combination of yeasts and LAB tested was observed. Yeast strains had no effect on LAB population density, except in three cases (i.e., pairs bB5-ham/yB10F-9, bB5-kim1/yB5-AC1, and bB5-kim6/yB6-15) where LAB population density decreased. A negative effect of *K. humillis* and *S. cerevisiae* on *L. sanfranciscensis* in dough was also recently observed [31].

In order to analyze the negative effect of LAB on yeast, acidity-sensitivity was measured for yeast strains. Even if the population size of *S. cerevisiae* strain (yB10-F9) in liquid media decreased at pH 3.5 and the population size of *K. humilis* strains decreased at pH 4 and below, dough pH was higher than 4 in more than 50% of tested doughs. Therefore, the negative effect of LAB on yeast growth could not be entirely explained by acidity-sensitivity of yeast. The threshold of acetate sensitivity was between 105 mM and 175 mM depending on the tested yeast strains. These levels were not reached in doughs in this study, in which the acetate concentration was also far over those reported in sourdoughs [12,32].

Negative effects of LAB on yeast were found even between yeast and LAB strains that were isolated from the same sourdough (sourdough B5). Therefore, LAB and yeast strains coexist naturally in sourdoughs, even if competition, amensalism, or some other type of negative interaction seem to be dominant. One explanation for the yeast/LAB coexistence, even in the presence of negative interactions, is that yeasts and LAB are unintentionally reinoculated regularly in sourdoughs because they are present in the bakery house environment. Indeed, a strong effect of the house microbiota on sourdough microbiota has been already described [16,24,32,33]. Another nonexclusive explanation for the coexistence of yeast and LAB in sourdoughs is that the sourdough production process never leads to the exhaustion of fermentable carbohydrates, and therefore, competition for resources is weak. In this study, after six hours of fermentation at 28 °C neither glucose, nor maltose, nor fructose were depleted, probably due to the continuous action of flour amylase. In practice, dough leavening lasts between three to twelve hours, but usually not longer than five hours. Even if longer fermentations are to be performed, they are processed partly at 4 °C, reducing the fermentation rate. Moreover, backslopping (i.e., reinoculation of a new batch of flour and water) is performed every day or so when the sourdough is not kept at a cold temperature [16,24,34]. Backslopping occurs with this frequency since bread is produced every day in bakeries, but also because after four days, exhaustion of carbohydrates does occur. In these bread-making scenarios, competition for carbohydrate resources may therefore not be strong enough to exclude yeasts or LAB from sourdough, since nutrients are replaced regularly and fermentation is sometimes slowed down by cold conditions. In this study, it cannot be confirmed whether the density of the yeast populations decreased in cocultures due to a lack of other resources not measured here (e.g., nitrogen, amino-acids, etc.), by inhibition due to the accumulation of toxic compounds, or by costly metabolism regulation changes. However, a description of the carbon sources, organic acids, and fermenting profiles in mono and cocultures of strains was done, and their relationship with the metabolic interaction of yeasts and LAB can be hypothesized.

The analysis of *S. cerevisiae* strain yB10-F9 and obligately heterofermentative LAB in monoculture confirm that they are maltose-positive strains. The *S. cerevisiae* strain depleted 36% of the initial maltose content after 6 h of fermentation. It is known that in anaerobiosis or in the presence of a sufficient amount of glucose, *S. cerevisiae* consumes glucose and fructose in preference of other mono or disaccharides. This is achieved by a glucose-induced repression of the genes involved in the catabolism of other carbohydrates [35,36,37,38]. In these experiments, the level of glucose in dough did not exceed 0.2% in monocultures, which was insufficient for glucose repression to occur. Indeed, glucose repression was initially shown to occur in *S. carlsbergensis* when cells were incubated with over 0.3% glucose [39]. We can therefore hypothesize that a lack of glucose repression, as well as maltose induction for the synthesis of maltose permease and alpha-glucosidase, leads to a decrease of maltose content in the dough at 6 h of fermentation. When inoculated alone, *S. cerevisiae* and obligately heterofermentative LAB strains increased the level of glucose in the dough. It is known that LAB transport maltose inside the cell and hydrolyze it by intracellular alpha-glucosidase into glucose. Sucrose can also be hydrolyzed into glucose and fructose inside and outside the cell [40]. *S. cerevisiae* alpha-glucosidase and invertase activities can also release glucose [41]. Taken together, these metabolic regulation processes could explain the increased glucose content observed in monocultures. Moreover, fructose levels were not increased after 6 h of fermentation, suggesting that fructose released by invertase activity was preferentially consumed over glucose, again in concordance to a lack of glucose-induced repression in monocultures.

Interestingly, in cocultures of *S. cerevisiae* and obligately heterofermentative LAB, glucose content was not increased, as observed in monocultures, but decreased. This shows that yeast/bacteria interactions alter the metabolic regulation observed in monoculture. The rate of maltose or sucrose hydrolysis by yeast may have been decreased in the presence of LAB. For instance, alpha-glucosidase and invertase activities are known to be sensitive to pH changes. Alternatively, LAB may have initially increased glucose levels by maltose hydrolysis, allowing yeast’s metabolic switch to glucose consumption to take place, and avoiding competition for maltose. Different levels of glucose repression can be found in *S. cerevisiae*. Leaky levels of catabolic repression in *S. cerevisiae* lead to a more general metabolic strategy that is costly for the cell but which allows a shorter lag-phase to occur when a switch to an alternative carbon source is needed quickly [42]. Sourdough is a variable environment where variation in carbohydrate content and resource input could have selected for generalist metabolic strategies. Competition for maltose between *S. cerevisiae* and the obligately heterofermentative LAB strain, as well as a cost associated with the metabolic reprogramming in yeast, would explain the yB10-F9 population density decrease in coculture. This might also explain why dough leavening was not increased and ethanol content was decreased in these cocultures compared to *S. cerevisiae* monoculture.

The cocultures of maltose-negative *K. humillis* strains with obligately heterofermentative maltose-positive LAB strains also revealed evidence of metabolic interaction. In maltose-negative *K. humilis* monocultures, the level of maltose increased compared to noninoculated doughs, suggesting that starch is actively degraded into maltose by yeast amylase. On the other hand, in obligately heterofermentative LAB monocultures, the level of maltose is not significantly decreased in dough compared to noninoculated doughs, suggesting that maltose consumption by obligately heterofermentative LAB strains is low, or is compensated for by the released of maltose from starch by flour amylases. Interestingly, in *K. humilis*/obligately heterofermentative LAB cocultures, the amount of maltose that is depleted is higher than in LAB monocultures. This suggest that cocultures facilitate the maltose uptake or/and hydrolysis, or that cocultures decrease the amylase activity, and thus, the release of maltose from starch. This was true only for *K. humilis*/obligately heterofermentative LAB pairs.

As mentioned, the interaction between *K. humilis* and obligately heterofermentative LAB strains has been usually described as being directed by cross-feeding. Indeed, cross-feeding, where obligately heterofermentative LAB cells hydrolyze maltose releasing glucose that becomes available for fermentation by maltose-negative *K. humillis*, is the most commonly described mechanism of interaction in sourdoughs [24,25,26]. In this study, doughs inoculated with obligately heterofermentative LAB contained significantly higher glucose concentrations than noninoculated doughs, indicating the release of glucose. However, increased glucose levels were also found when maltose-negative *K. humilis* strains were cultivated with obligately heterofermentative LAB compared to yeast monocultures. Therefore, there was no clear evidence that yeast strains consume the glucose produced by the obligately heterofermentative LAB. Because more maltose was depleted in cocultures than in monocultures, it could be hypothesized that more glucose was released by maltose positive bacteria in cocultures, and then consumed by yeasts. Alternatively, maltose may have been transformed into organic acids, CO_2_, and ethanol by LAB. Additional experiments in synthetic media should be done to understand the mechanisms behind these metabolic interactions.

Even if positive interactions are hypothesized to rule out the coexistence of microbial species, negative interactions, particularly competition, have already been described as the most common type of interaction between pairs of microbial species in microbial communities. Indeed, there is growing evidence showing that the typical result of microbial evolution in communities will be competitive, rather than cooperative [43,44]. Recent studies have even predicted that competition improves microbial community stability and drives community assembly [45]. The role of competition for resources in microbial interactions has been widely described, but many questions still prevail [46,47]. Food microorganism interactions are not excluded from this, and this work contributes to showing that sourdough communities, while simple, also harbor complex interactions. Further analysis should be done to establish the real importance of the nature of these interactions in the stability of these communities. Moreover, it is worth noting that sourdoughs, but also many other food ecosystems, are good models by which to study microbial interactions in environments where no resource exhaustion is reached and species extinction may be unlikely.

## Figures and Tables

**Figure 1 microorganisms-08-00240-f001:**
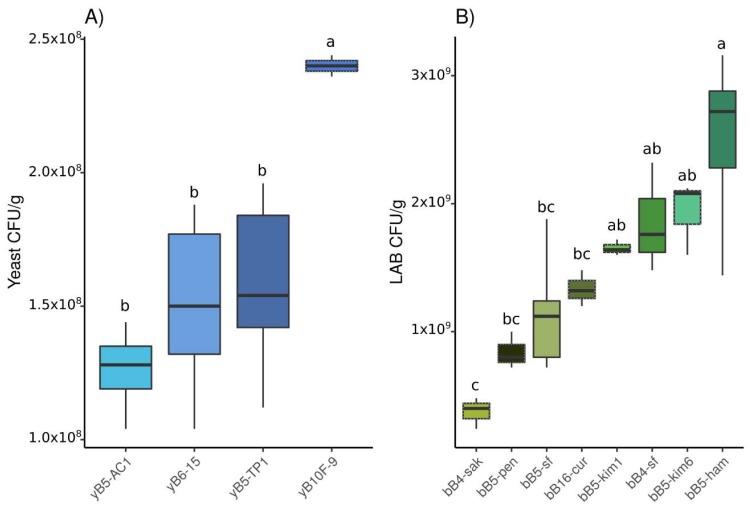
Colony forming units per gram of dough (CFU/g) of (**A**) yeast strains in monoculture and (**B**) LAB strains in monoculture after 24 h. Compact letter display on top indicate Tuckey’s statistical groups. Blue was chosen for yeast strains; dashed lines indicate *S. cerevisiae* and plain lines *K. humilis*. Green was chosen for LAB strains, with obligately heterofermentative LAB in plain lines boxplots.

**Figure 2 microorganisms-08-00240-f002:**
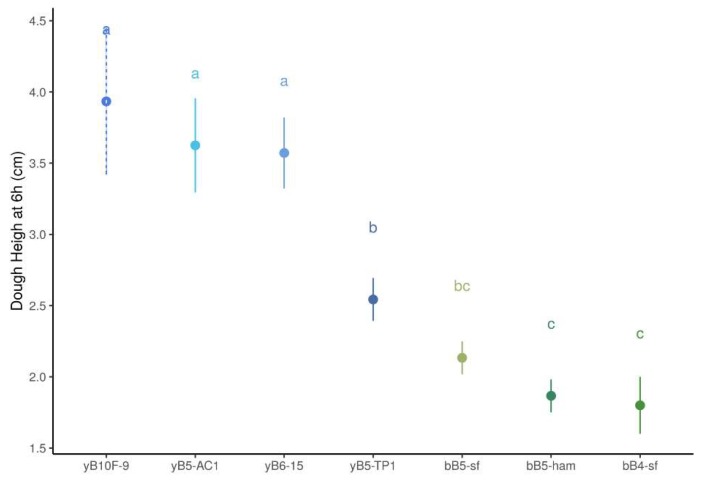
Dough height for monocultures at 6 h for yeast strains in blue (dashed lines for *S. cerevisiae* and plain lines for *K. humilis*) and obligately heterofermentative LAB strains in green. Compact letter display on top indicate Tuckey’s statistical groups.

**Figure 3 microorganisms-08-00240-f003:**
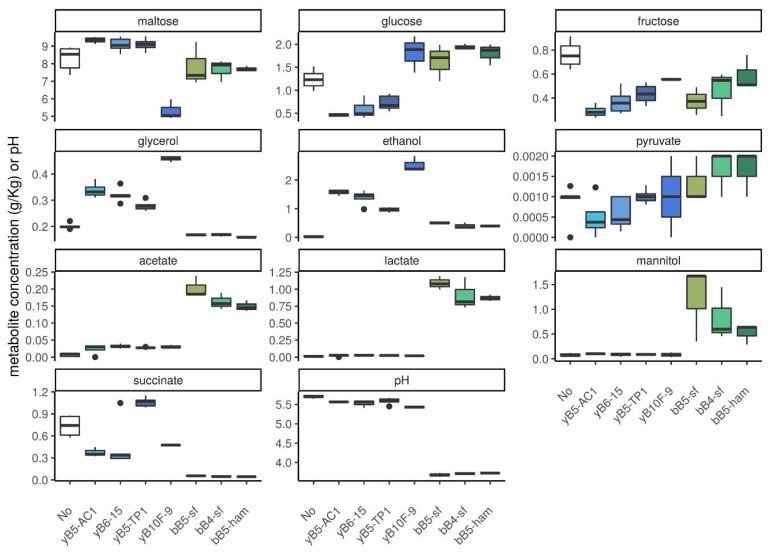
Metabolite concentration and pH in monocultures of yeasts and obligately heterofermentative LAB strains. Green indicates obligately heterofermentative LAB. Blue indicates yeasts, with *S. cerevisiae* in dashed lines and *K. humilis* in plain lines.

**Figure 4 microorganisms-08-00240-f004:**
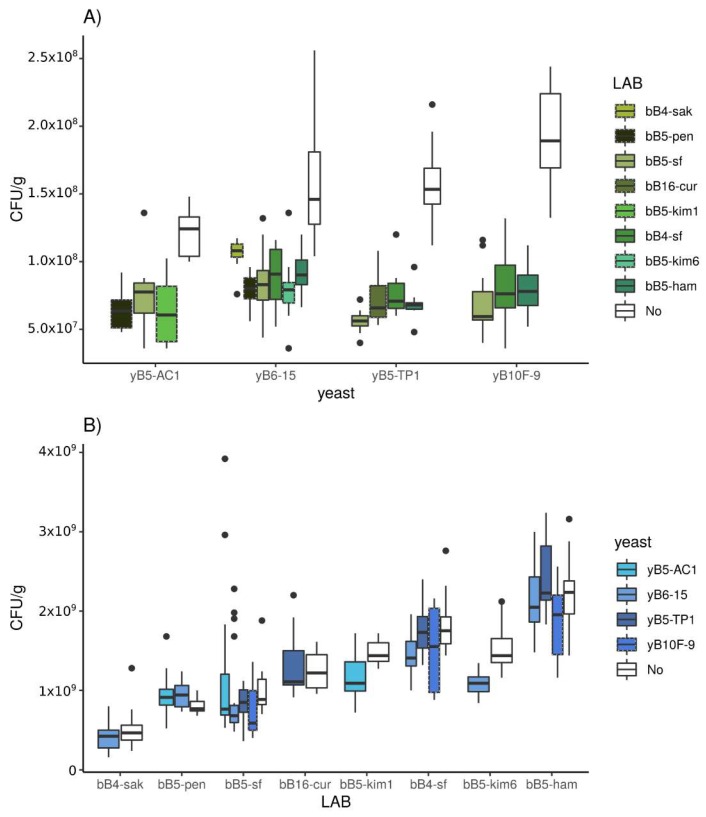
Colony forming units per gram of dough (CFU/g) for (**A**) yeast strains in monoculture and coculture with LAB and (**B**) LAB strains in monoculture and coculture with yeast, measured after 24 h. CFUs/g in monoculture are presented in white-filled boxes. Green indicates LAB, with obligately heterofermentative LAB strains in a plain lined boxplot. Blue indicates yeast, with *S. cerevisiae* in dashed boxplots and *K. humilis* in a plain lined boxplot.

**Figure 5 microorganisms-08-00240-f005:**
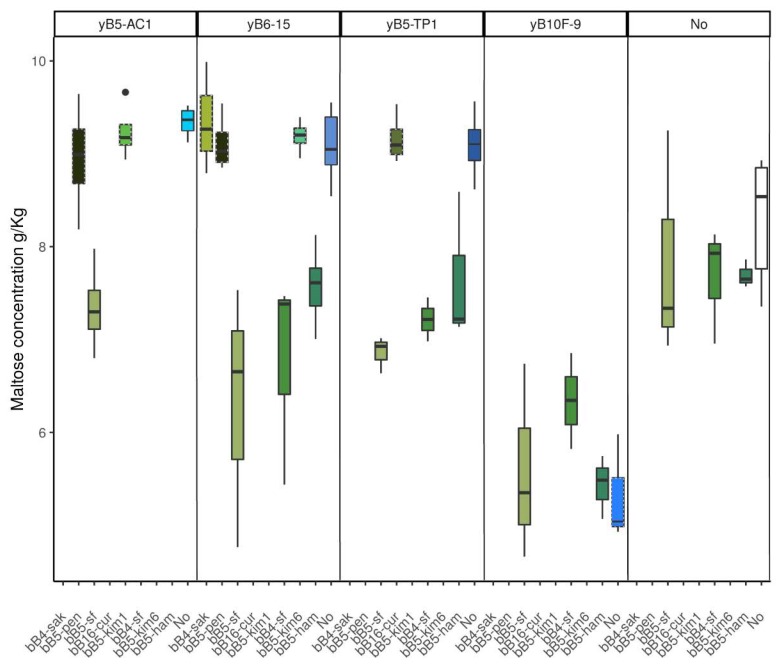
Maltose concentration after 6 h of fermentation. Each panel indicates fermentation for one of the tested yeast strains in coculture (green boxes) or monoculture (blue boxes). Obligately heterofermentative LAB strains are represented in plain lined boxes. *S. cerevisiae* is represented in dashed lined boxes while *K. humilis* in plain lined boxes. White boxes represent control dough without inoculum.

**Figure 6 microorganisms-08-00240-f006:**
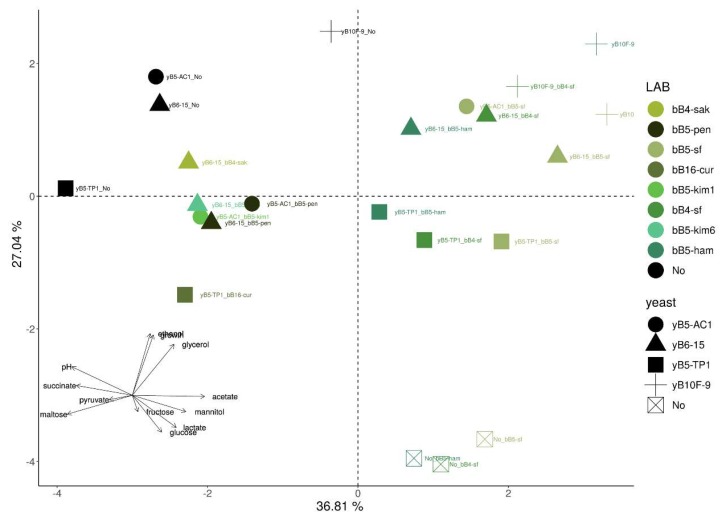
Principal component analysis based on average metabolite levels, pH, and dough height of fermentation. LAB strains are represented with different colors and yeast strains with different symbols. Arrows represent a graphical representation of the loadings of each original variable.

**Table 1 microorganisms-08-00240-t001:** Collection of strains.

Species	Strain’s Code	Lab Code	Baker	French Region	Main Physiological Characteristic
Yeast					
*Saccharomyces cerevisiae*	yB10F-9	MTF3947	B10	Provence-Alpes Côte d’Azur	maltose-positive
*Kazachstania humilis*	yB6-15	MTF3948	B6	Champagne Ardennes	maltose-negative
*Kazachstania humilis*	yB5-TP1	MTF3949	B5	Ile de France	maltose-negative
*Kazachstania humilis*	yB5-AC1	MTF4070	B5	Ile de France	maltose-negative
**Bacteria**					
*Lactobacilus curvatus*	bB16-cur	MTF 4123	B16	Ile de France	facultatively heterofermentative
*Lactobacilus sakei*	bB4-sak	MTF 4118	B4	Pays de Loire	facultatively heterofermentative
*Lactobacilus sanfranciscensis*	bB4-sf	MTF 3946	B4	Pays de Loire	obligately heterofermentative
*Lactobacilus sanfranciscensis*	bB5-sf	MTF 3945	B5	Ile de France	obligately heterofermentative
*Lactobacilus hammesii*	bB5-ham	MTF 3944	B5	Ile de France	obligately heterofermentative
*Lactobacilus pentosus*	bB5-pen	MTF 4114	B5	Ile de France	facultatively heterofermentative
*Lactobacilus kimchi*	bB5-kim1	MTF 4117	B5	Ile de France	facultatively heterofermentative
*Lactobacilus kimchi*	bB5-kim6	MTF 4116	B5	Ile de France	facultatively heterofermentative

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
