# Peer review of "Interactions between Kazachstania humilis Yeast Species and Lactic Acid Bacteria in Sourdough"

_microorganisms, 2020, doi:10.3390/microorganisms8020240_

Round 1

Reviewer 1 Report

This manuscript examines the postulated hypothesis of the mutualistic interaction between maltose-negative Kazachstania humilis yeast strains and maltose-positive lactic acid bacteria strains. As the authors try to disapprove this hypothesis, they should have measured as much parameters as possible to proof this. A major concern is the fact that the fate of fructose is not fully examined, as mannitol was not measured. The hypothesis says that yeasts elaborate fructose from fructans and that fructose is reduced to mannitol by strictly heterofermentative LAB strains possessing maltose phosphorylase activity to enhance their competitiveness. This has not been shown in the present manuscript and hence the conclusion should be weakened by addressing this point. Also succinic acid, as a measure for yeast activity, could have been measured. Finally, to confirm the acid-sensitivity of the yeast strains used, they should have been tested for acid tolerance.

Lines 8-9, 44-45, and 392-393: this reviewer does not agree with this statement. See meta-analysis study of Van Kerrebroeck et al., 2017, Trends in Food Science and Technology 68, 152-159.

Line 13 and throughout the manuscript: avoid to use personalized writing; use the passive mode.

Line 16: replace the word ‘famous’ by ‘existing’ or ‘postulated’.

Line 17 and throughout the manuscript: this statement is too general; specify the LAB species.

Line 21 and throughout the manuscript: write ‘maltose-positive’ and ‘maltose-negative’ with a hyphen.

Line 23, 48: Latin names of microorganisms should be italic.

Line 32 and throughout the manuscript: write ‘yeasts’ (plural).

Line 34, 91: write the chemical formula of carbon dioxide correctly.

Line 38: write ‘fermented food products’.

Line 43: write ‘leavening’.

Line 45 and throughout the manuscript: write ‘prevailing’ instead of ‘dominant’.

Line 46: write ‘when considering different sourdoughs’ instead of ‘in different sourdoughs’.

Line 47: add a sentence that states that, in general, one or two yeast species and less than three different LAB species inhabit a single bakery sourdough (Van Kerrebroeck et al., 2017).

Line 63: write ‘are also called’.

Line 68: write ‘is taken up’.

Line 69: write ‘by a dedicated maltose phosphorylase’.

Line 70: write ‘mainly under stress conditions’.

Line 74: write ‘LAB growth’ instead of ‘LAB cells growth’.

Line 98 and throughout the manuscript: write ‘carbohydates’ or ‘saccharides’ instead of ‘sugars’.

Line 99: why was mannitol not measured? Mannitol is a good indicator of heterofermentative LAB growth in the presence of fructose as alternative external electron acceptor.

Line 111: write ‘were’ instead of ‘was’.

Line 112: write strains’ names.

Line 118 and throughout the manuscript: use the International Unit abbreviation ‘h’ for ‘hour’, ‘hours’ or ‘hs’.

Line 126 and throughout the manuscript: use the International Unit abbreviation ‘g’ for ‘gs’.

Line 145 and throughout the manuscript: write ‘counts’ instead of ‘count’.

Line 147 and throughout the manuscript: use the International Unit abbreviation ‘min’ for ‘minutes’.

Line 149: write ‘plated on’.

Lines 160, 162: write chemical formulas correctly.

Line 187: write ‘detected’.

Line 206: write ‘bacterial strain’.

Lines 213, 320, 362, and 431: write ‘mono- and co-cultures’ (with hyphen).

Line 231: write ‘FDR’ in full.

Line 243 and throughout the manuscript: write ‘on average’.

Line 272, 278: write ‘organic acids’.

Line 285/286: how do you explain the decrease in fructose concentration in yeast monocultures?

Line 295: how do you explain the decrease in glycerol concentrations in LAB monocultures?

Line 352: write ‘lactic acid production’.

Line 353: write ‘did not show’.

Line 358-362: is this not a proof for the postulated hypothesis of the interaction between maltose-negative K. humilis and maltose-positive, strictly heterofermentative LAB strains?

Line 370: spell ‘represented’ correctly.

Lines 394/395: this hypothesis is not general but focusses on maltose-negative K. humilis and maltose-positive Lb. sanfranciscensis.

Line 401: too general; see comments above (line 17 and lines 394/395).

Line 422: backslopping for more than four days often results in exhaustion of the carbohydrates. Please reformulate.

Line 424: write ‘processed’.

Line 427: write ‘slowed down’.

Line 428: do you mean ‘carbohydrate substrates’?

Line 437: write ‘mono- or disaccharides’ (with hypen and in plural).

Lines 445-447: this is too general; differences exist among lactobacilli, certainly in a sourdough ecosystem. Please adapt and rephrase.

Lines 449-451: what about fructan degradation by yeasts? See comment above.

Line 446 and further: here, this reviewer has the impression that the authors do acknowledge the mutualistic interaction between maltose-negative K. humilis and maltose-positive, strictly heterofermentative lactobacilli. This is confusing when reading the manuscript. Therefore, as mentioned above, distinction should be made between the presumed mutualistic interaction (based on a trophic relationship) between maltose-negative K. humilis and maltose-positive, strictly heterofermentative Lb. sanfranciscensis; potential interactions between maltose-negative K. humilis and other maltose-positive, strictly heterofermentative LAB species; and potential interactions between other maltose-negative or maltose-positive yeast species and other maltose-negative or maltose-positive, strictly heterofermentative, facultatively heterofermentative, or homofermentative LAB species. This distinction should be made clearly throughout the whole manuscript.

Author Response

We thank the Reviewer 1 for his/her feedback and we appreciate the comments and suggestions for improving our work. As suggested we have added mannitol and succinate quantifications. We have as well tested the acid-sensitivity of the yeast strains and also measured pH of doughs at 24h. We believe that adding these results have improved our manuscript, thanks.

Concerning fructose fate, we have observed that doughs inoculated with yeast mono-cultures always present lower fructose concentration than non-inoculated control doughs or than the doughs inoculated with LAB mono-cultures. Therefore, we were not able to find evidence that fructose was released from fructans by yeast. However, as expected, we did find that obligately LAB produced mannitol.

Succinate is present in low amounts in our experimental units. There is an initial amount of succinate present in non-inoculated doughs which makes the interpretation about yeast activity difficult. The lowest levels of succinate were found in LAB mono-cultures. K. humilis yeast strain yB5−TP1 showed the highest levels of succinate while in mono-culture.

Concerning yeast acid-sensitivity, we found that K. humilis  strains are more sensitive to pH decrease but more tolerant to acetate increase than the tested S. cerevisiae strain. However, over half of the dough units inoculated with co-cultures showed pH over the threshold detected for K. humilis and S. cerevisiae. Therefore, low pH cannot be the sole explanation of the negative impact of LAB on yeasts.

We have discussed these results in the manuscript.

Lines 8-9, 44-45, and 392-393: this reviewer does not agree with this statement. See meta-analysis study of Van Kerrebroeck et al., 2017, Trends in Food Science and Technology 68, 152-159.

We have changed the statement according to Van Kerrebroeck et al., 2017 and added the citation.

Line 13 and throughout the manuscript: avoid to use personalized writing; use the passive mode.

We have made the appropriate style modifications.

Line 16: replace the word ‘famous’ by ‘existing’ or ‘postulated’.

We have replaced the term as suggested.

Line 17 and throughout the manuscript: this statement is too general; specify the LAB species.

We have specified the LAB species and their metabolism throughout the manuscript.

Line 21 and throughout the manuscript: write ‘maltose-positive’ and ‘maltose-negative’ with a hyphen.

We have replaced the term as suggested.

Line 23, 48: Latin names of microorganisms should be italic.

We have corrected it.

Line 32 and throughout the manuscript: write ‘yeasts’ (plural).

We have replaced the term as suggested.

Line 34, 91: write the chemical formula of carbon dioxide correctly.

We have corrected it.

Line 38: write ‘fermented food products’.

We have replaced the term as suggested.

Line 43: write ‘leavening’

We have replaced the term as suggested

Line 45 and throughout the manuscript: write ‘prevailing’ instead of ‘dominant’.

We have replaced it as suggested

Line 46: write ‘when considering different sourdoughs’ instead of ‘in different sourdoughs’.

We have replaced it as suggested

Line 47: add a sentence that states that, in general, one or two yeast species and less than three different LAB species inhabit a single bakery sourdough (Van Kerrebroeck et al., 2017).

We have added the sentence as suggested.

Line 63: write ‘are also called’.

We have corrected it.

Line 68: write ‘is taken up’.

We have corrected it.

Line 69: write ‘by a dedicated maltose phosphorylase’

We have replaced it as suggested

Line 70: write ‘mainly under stress conditions’.

We have added the phrase as suggested

Line 74: write ‘LAB growth’ instead of ‘LAB cells growth’.

We have replaced it as suggested

Line 98 and throughout the manuscript: write ‘carbohydates’ or ‘saccharides’ instead of ‘sugars’.

We have replaced it as suggested

Line 99: why was mannitol not measured? Mannitol is a good indicator of heterofermentative LAB growth in the presence of fructose as alternative external electron acceptor.

Mannitol levels were quantified as requested, results were added to the manuscript

Line 111: write ‘were’ instead of ‘was’.

We have replaced it as suggested

Line 112: write strains’ names.

We have replaced it as suggested

Line 118 and throughout the manuscript: use the International Unit abbreviation ‘h’ for ‘hour’, ‘hours’ or ‘hs’.

We have replaced it as suggested

Line 126 and throughout the manuscript: use the International Unit abbreviation ‘g’ for ‘gs’

We have corrected this as suggested

Line 145 and throughout the manuscript: write ‘counts’ instead of ‘count’.

We have replaced it as suggested

Line 147 and throughout the manuscript: use the International Unit abbreviation ‘min’ for ‘minutes’.

We have replaced it as suggested

Line 149: write ‘plated on’.

We have replaced it as suggested

Lines 160, 162: write chemical formulas correctly.

We have corrected this as suggested

Line 187: write ‘detected’.

We have corrected this as suggested

Line 206: write ‘bacterial strain’

We have corrected this as suggested

Lines 213, 320, 362, and 431: write ‘mono- and co-cultures’ (with hyphen)

We have corrected this as suggested

Line 231: write ‘FDR’ in full

We have corrected this as suggested

Line 243 and throughout the manuscript: write ‘on average’.

We have replaced it as suggested

Line 272, 278: write ‘organic acids’

We have replaced it as suggested

Line 285/286: how do you explain the decrease in fructose concentration in yeast monocultures

We believe that even if glucose is not depleted from dough at 6h, its concentration does decrease, probably yeast will switch to fructose consumption in low glucose concentration. As we have stated in the text “ethanol concentration was significantly correlated to the predicted concentration of ethanol molecules produced from glucose and fructose”.

Line 295: how do you explain the decrease in glycerol concentrations in LAB monocultures?

We don’t have any explanation. The strains used have been characterized on API50CH (shown on Table S1 now) and none of them were able to assimilate glycerol. If you have any hypothesis, we will be grateful if you could help us to understand this result.  

Line 352: write ‘lactic acid production’.

We have corrected it as suggested

Line 353: write ‘did not show’

We have corrected it as suggested

Line 358-362: is this not a proof for the postulated hypothesis of the interaction between maltose-negative K. humilis and maltose-positive, strictly heterofermentative LAB strains?

We rewrote the discussion on that point to make it more clear.

We found that K. humilis increased the level of maltose in dough, probably thanks to its amylase activity. When adding strictly heterofermentative LAB to K. humilis, the maltose concentration in dough decreased to a level that is closed to the maltose level in LAB monoculture. Two hypothesis can explain these results:

-either LABs reduce/inhibit the activity of yeast amylase by reduicing the pH

-or/and LAB assimilates more maltose in presence of yeast than alone

Increased glucose levels were found in strictly heterofermentaive LAB monocultures compared to non-inoculated dough indicating that LABs release glucose. However, similar increase of glucose was found when maltose-negative K. humilis strains were cultivated with obligately heterofermentative LAB. Therefore, there was no clear evidence that yeast strains consume the glucose produced by the obligately heterofermentative LAB.

Line 370: spell ‘represented’ correctly.

We have corrected it as suggested

Lines 394/395: this hypothesis is not general but focusses on maltose-negative K. humilis and maltose-positive Lb. sanfranciscensis.

We have re-phrased it to make it more clear.

Line 401: too general; see comments above (line 17 and lines 394/395).

We have specified the LAB species and their metabolism throughout the manuscript.

Line 422: backslopping for more than four days often results in exhaustion of the carbohydrates. Please reformulate.

We have corrected it as suggested

Line 424: write ‘processed’.

We have corrected it as suggested

Line 427: write ‘slowed down’

We have replaced it as suggested

Line 428: do you mean ‘carbohydrate substrates’?

Yes we did

Line 437: write ‘mono- or disaccharides’ (with hypen and in plural).

We have replaced it as suggested

Lines 445-447: this is too general; differences exist among lactobacilli, certainly in a sourdough ecosystem. Please adapt and rephrase.

We have rephrased and have specified the LAB species and their metabolism throughout the manuscript.

Lines 449-451: what about fructan degradation by yeasts? See comment above.

As mentioned above, we have observed that doughs inoculated with yeast mono-cultures always present lower fructose concentration than non-inoculated control doughs or than the doughs inoculated with LAB mono-cultures. Therefore, we were not able to find evidence that fructose was released from fructans by yeast.

Line 446 and further: here, this reviewer has the impression that the authors do acknowledge the mutualistic interaction between maltose-negative K. humilis and maltose-positive, strictly heterofermentative lactobacilli. This is confusing when reading the manuscript. Therefore, as mentioned above, distinction should be made between the presumed mutualistic interaction (based on a trophic relationship) between maltose-negative K. humilis and maltose-positive, strictly heterofermentative Lb. sanfranciscensis; potential interactions between maltose-negative K. humilis and other maltose-positive, strictly heterofermentative LAB species; and potential interactions between other maltose-negative or maltose-positive yeast species and other maltose-negative or maltose-positive, strictly heterofermentative, facultatively heterofermentative, or homofermentative LAB species. This distinction should be made clearly throughout the whole manuscript.

We have corrected the manuscript in an effort to make this clearer. We, however, still stand that negative ecological outcomes are observed in our experiment for all pairs of yeast and bacteria tested (yeast population density was always decreased by LAB). We also explain that metabolic interactions occur between maltose-positive, strictly heterofermentative LAB strains and maltose-negative K. humilis strains but that our results do not give any clear evidence that yeast consumes the glucose produced by LAB.

Reviewer 2 Report

The authors presented a study on a hypothetical mutualism between the strains of Kazachstania humilis and some lactic acid bacteria isolated from sourdoughs. However, they do not indicate the criteria for choosing the strains used. it would be advisable to report a table showing the name of the species of yeasts and bacteria used and their main physiological characteristics that led to their choice. The experimentation does not bring new scientific knowledge on the physiological and metabolic interactions between the various microbial populations in  sourdoughs. It would be appropriate to investigate more on the metabolism of the various microbial populations that colonize sourdoughs and that bring and justify their coexistence.

Author Response

We thank Reviewer 2 for his comments. We have included a table showing the name of the species of yeasts and bacteria used, the strains´names, the place of origin and their main metabolic characteristics. We have also included two additional sup. tables giving a physiological characterization of each strain. For each LAB and yeast strain, the assimilation pattern of 50 and 30 substrates, respectively, is described.

We believe there is novelty in our work since K. humilis strains have been poorly studied, while LAB have been deeply studied. We have also added to the new version of the manuscript results on acid-sensitivity of the yeast strains used. This adds novel knowledge on sourdough yeast diversity. We have also revealed different fermentation and metabolic patterns between K. humilis strains.

The main aim of our work was to analyze the ecological nature of the yeast/bacteria interactions in sourdough environments. To our knowledge, we have shown for the first time that yeast growth can be hampered by the presence of LAB in dough. The main metabolic products are measured in this paper in order to add information to the metabolic outcomes of the interactions. We did not aim to bring new scientific knowledge on the metabolic interaction of each tested strains, but to get deeper insight into their ecological relationships in dough.  The analysis of metabolic interaction between yeast and LAB strains would have required a deep study in laboratory media but would not have given information on the metabolic outcomes in dough, as dough is a complex ecosystem. Our main final conclusion is that interactions between strictly heterofermentative LAB strains and maltose-negative K. humilis strains are not ecologicaly positive, at least not for K. humilis strains, this would be at least a case of amensalism.

Round 2

Reviewer 1 Report

Personalized forms should be replaced throughout the whole manuscript.

Reviewer 2 Report

The revisions made make the part of the materials and methods acceptable. Concerns remain about the part of the conclusions of the work. The authors say verbatim: "as dough is a complex ecosystem". Precisely for this reason, future insights on the metabolic interactions that occur "in vivo" appear crucial and fundamental. the results of these investigations would help to better understand how a microorganism, in a complex matrix can be considered "pro-technological" or on the contrary "unwanted".
The "in vitro" results highlight the interactions between K. humilis and heterofermentative Labs cannot consider positive but future in vitro and in vivo investigations can help us understand if positive interactions between these 2 microbial groups, associated with other physiological characteristics and metabolic (e.g. production of prebiotic substances that can generate symbiotic effects).